# Revisiting the functional significance of binocular cues for perceiving motion-in-depth

Peter J. Kohler [1], Wesley J. Meredith [1] & Anthony M. Norcia [1]

Binocular differencing of spatial cues required for perceiving depth relationships is associated with decreased sensitivity to the corresponding retinal image displacements. However, binocular summation of contrast signals increases sensitivity. Here, we investigated this divergence in sensitivity by making direct neural measurements of responses to supra-threshold motion in human adults and 5-month-old infants using steady-state visually evoked potentials. Interocular differences in retinal image motion generated suppressed response functions and correspondingly elevated perceptual thresholds compared to motion matched between the two eyes. This suppression was of equal strength for horizontal and vertical motion and therefore not specific to the perception of motion-in-depth. Suppression is strongly dependent on the presence of spatial references in the image and highly immature in infants. Suppression appears to be the manifestation of a succession of spatial and interocular opponency operations that occur at an intermediate processing stage either before or in parallel with the extraction of motion-in-depth.

[1] Department of Psychology, Stanford University, Stanford, CA 94305, USA. Correspondence and requests for materials should be addressed to P.J.K. (email: pjkohler@stanford.edu)

There is robust sensitivity to both direction of motion and retinal disparity in primary and higher-order visual cortex of primates. Direction tuning is present within the classical receptive field[1–4], but can be modified by motion in the surround. These surround effects[5–7] convey sensitivity to motion-defined discontinuities and to relative motion (where two or more velocities can be compared). Psychophysically, human observers are much more sensitive to relative motion than to absolute (unreferenced) motion[8–10].

Disparity tuning is also strongly present in V1 classical receptive fields[11–13]. Unlike the case just described for motion, the disparity of stimuli in the non-classical surround has little or no effect on V1 disparity tuning[14,15]. By contrast, in V2 and beyond responses to disparate stimuli within the classical receptive field do depend on the disparities present in the surround, to varying degree[16–19]. As for motion, psychophysical measurements indicate that human observers are much more sensitive to relative disparity than absolute (unreferenced) disparity[20,21].

Motion responses in V1 and V2 and other visual areas such as the middle temporal area have been studied primarily with stimuli that move in the fronto-parallel plane—somewhat of a special case, since objects move in three dimensions. With natural stimuli, there are two main cues that can be decoded to signal three-dimensional motion-in-depth (MID). The visual system can read out the binocular disparity of an object relative to the fixation plane and track how this information varies over time (change of disparity over time or CDOT). Another possibility is to compare object velocity from each monocular image (interocular velocity difference or IOVD). Both cues provide partial but not complete information for MID[22]. Psychophysical thresholds are higher for fronto-parallel motion than for MID, suggesting a fundamental difference in how MID is handled by the visual system[23–26].

Our approach builds on these results and prior work with the visually evoked potential (VEP) on motion processing[27,28] and spatial integration[29–31] to systematically explore neural responses to motion and disparity using minimally complex scene structures containing regions defined by motion, disparity, or both. VEPs provide direct neural measures that can bridge the near-threshold regime used in the psychophysical literature and the suprathreshold regime used in the primate electrophysiology literature, allowing us to test previously proposed computational mechanisms for binocular motion processing[24,25,32]. Our use of steady-state VEPs (SSVEPs) provides an a priori means of interpreting response components in the frequency domain based on response symmetry considerations rather than on the unconstrained basis of peak amplitudes and latencies as common in time-domain approaches[33]. Critically, we take advantage of the fundamental asymmetry in retinal stimulation caused by the lateral separations of the eyes and compare stimuli with horizontally displaced motion signals that are ecologically relevant to MID, and to stimuli with vertically displaced motion signals that are not. This allows us to separate neural responses specifically adapted to MID from those that support more generic image-processing functions. In a subset of our experiments, we collect both neural and psychophysical data at the same time, and directly relate brain responses to perception.

Prior work in the developing human visual system using VEPs has shown that while responsiveness to absolute disparity is robust by 4–6 months of age[34], relative disparity processing mechanisms are strikingly immature at this age[35]. Moreover, both human and non-human primate motion processing mechanisms are immature during infancy and depend on the presence of normal eye alignment during early development, implicating specifically binocular motion processing mechanisms[36–40]. Given these prior results, it was natural to address visual developmental status during infancy with binocular stimuli containing both motion and disparity cues for MID.

We identify a motion-related response component that exhibits suppression when interocular differences in retinal image motion are present and is strongly dependent on references, but does not depend on whether the stimulus is compatible with MID. This component appears to be driven by IOVD and is highly immature in 5-month-old infants. We also identify a second component that has a similar response profile, but is more sensitive to displacements under MID-compatible conditions. Taken together, our results show that both IOVD and CDOT are extracted either before or in parallel with activity that is relevant for the perception of MID.

## Results

**Overall approach.** The experimental stimuli are illustrated schematically in Fig. 1. The logic of the experimental design was to measure responses that are phase synchronized to the frequency of monocular apparent motion (2 Hz), but that are modulated with respect to interocular phase relationships, interocular image correlation, or the availability of reference cues for motion or disparity. In addition, we varied the orientation of the display, which could be either horizontal or vertical. Each factor contributes to the binocular interpretation of the stimuli, but leaves the monocular motion information unchanged across conditions. Responses that differ as a function of the interocular phase are specifically binocular because interocular phase is only available after the point of binocular convergence. Responses that differ as function of the interocular correlation also reflect specifically binocular mechanisms. Responses that depend on the orientation of the display will reflect mechanisms that are specifically adapted to MID, which can only be computed from the horizontal displays. Responses that do not depend on orientation, but do depend on interocular phase and correlation are binocular, but not MID specific. Finally, responses that depend on the presence of a non-moving reference are evidence of mechanisms that integrate relational information across space.

In each experimental condition, dots underwent apparent motion at 2 Hz and the amplitude of the displacement was swept from 0.5 to 16 arcmin in 10 equal log steps (2–32 arcmin for infants), presented sequentially within a 10-s trial. The display spanned 40 × 40° and contained 20 moving test bands and 20 non-moving reference bands on the screen (spatial frequency of 0.5 c/deg). By systematically comparing the SSVEP amplitude as a function of monocular displacement over a series of five experiments, we could evaluate how relative motion and disparity cues, interocular phase, and the CDOT and IOVD cues determine the properties of the displacement response function. In addition to the five main experiments, we also present two psychophysical experiments, and an experiment with 5-month-old infant participants.

Frequency analysis was used to quantify the evoked responses generated by the 2 Hz visual stimuli. Periodic visual stimulation leads to an SSVEP, a periodic evoked response occurring at exact integer multiples of the stimulation frequency[33]. When a stimulus attribute is symmetric, say in terms of leftward and rightward motion, the SSVEP will be dominated by even harmonics of the stimulus frequency. When the stimulus configuration is asymmetric, for example, when it modulates between two different global configurations such as uniform and segmented, the SSVEP will also contain odd harmonics of the stimulus frequency. We first describe responses at the second harmonic that we attribute to the motion component of the stimuli. This is followed by a description of the results at the first harmonic that we attribute to the processing of the global structure and segmentation of the stimuli.

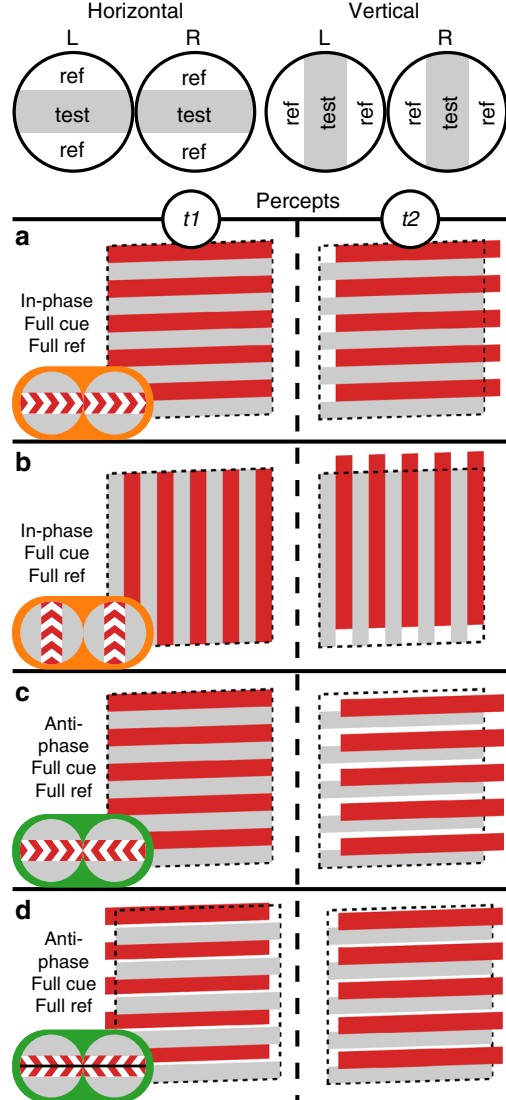

**Fig. 1** Schematic of the stimulus configurations. The displays were random-dot kinematograms with alternating test and reference bands (20 of each in the actual display, 5 shown here). Dots in test bands moved coherently either in-phase between the two eyes (**a**, **b**) or in anti-phase (**c**, **d**). In the reference bands, dots were either static, uncorrelated between the eyes, or removed completely, depending on the experiment and condition (see Fig. 2). Here we show FOUR conditions, each with static reference dots and each represented with an icon depicting the moving test bands as a central band with flanking reference bands. An illustration of the corresponding percept is shown next to each icon: Both horizontal (**a**) and vertical (**b**) in-phase motion gives rise to movement in the plane, while horizontal anti-phase motion means that the display will alternate between 0 and crossed horizontal disparity, giving rise to movement in depth (**c**). Vertical anti-phase motion alternates between 0 and left-hypo disparity, and does not give rise to movement in depth. The endpoints of the monocular apparent motion trajectories can be manipulated such that the horizontal motion will generate a display that alternates between equal and opposite values of crossed and uncrossed disparity, still giving rise to movement in depth (**d**). The vertical equivalent of this display alternated between left-hyper and left-hypo disparity, not giving rise to movement in depth. In addition to these schematics, videos of the actual displays are included as Supplementary Movies 1–4

**References are crucial for both in-phase motion and anti-phase motion.** The SSVEP related to both absolute (unreferenced) and relative (referenced) in-phase motion occurs at even harmonics of the stimulus frequency, with the second harmonic response

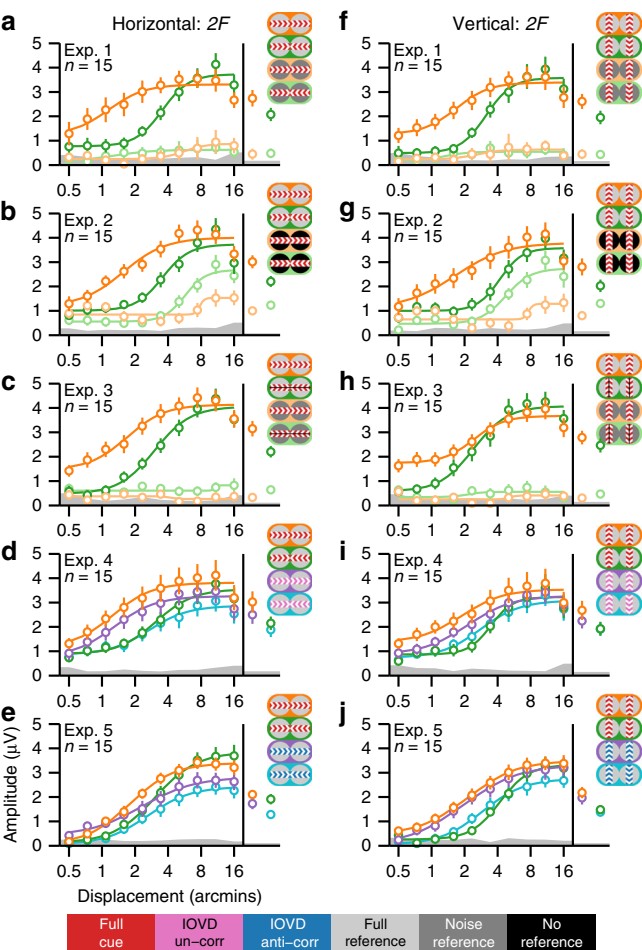

**Fig. 2** Adult second harmonic SSVEP response functions. **a–e** Displacement response functions for the horizontal direction of motion and **f–j** the vertical direction of motion. Averages across all displacements are shown on the right side of the response functions. Data are from the first reliable component from an RC analysis derived from 2F data from all conditions, separately for each experiment. Scalp topographies for this component were quite similar across experiments (see Supplementary Fig. 2). Each experiment had eight conditions, of which half were horizontal and the rest were vertical. The conditions are represented with icons, consistent with those used in Fig. 1. The central colors indicate the interocular correlation of the test bands, which could be either 100% (full-cue; red), 0% (IOVD-uncorrelated; magenta), or −100% (IOVD anti-correlated; blue). The flanking colors indicate the state of the reference bands, which either contained static dots that were 100% correlated between eyes (full-reference; light gray), temporally and interocularly uncorrelated dots (noise-reference; dark gray), or no dots (no-reference; black). The meaning of each color is shown in the legend in the bottom of the figure. Smooth curves are Naka–Rushton function fits to the data. The gray bands at the bottom of the plots indicate the background EEG noise level, with the top of the band indicating the average noise level across two neighboring side bands, averaged across conditions. Error bars plot ±1 standard error of the mean (SEM)

(2F/4 Hz) being the largest (see Fig. 2 for 2F and Supplementary Fig. 1 for 4F/8 Hz response functions).

In the first condition of Experiment 1, the test bands were moving in-phase and were flanked by reference bands containing static dots (full-cue/full-reference; Fig. 1a), and the SSVEP amplitude is a saturating function of horizontal displacement (Fig. 2a, dark orange). This response function is well described by

the Naka–Rushton function[41] and fits of this function to the data are plotted as an aid to visualization in Fig. 2 and elsewhere. In the second condition of Experiment 1, the reference dots were temporally and interocularly uncorrelated (noise-reference), making it impossible to calculate a unique relative velocity because the reference bands contain a very broad and random distribution of velocities. This manipulation strongly reduces the response amplitude for in-phase motion (Fig. 2a, light orange). In the vertical direction, in-phase motion produced very similar *2F* responses, with similar differences between full-reference and noise-reference conditions (Fig. 2f).

The noise reference in Experiment 1 may have masked the moving dots through suppressive lateral interactions. We tested for this in Experiment 2 by using a display in which the moving bands were the same as in Experiment 1 but the reference bands were blank (no-reference). We also included the full-reference conditions of Experiment 1 for a within-observer comparison. The no-reference conditions produced stronger responses than noise-reference conditions, but a large difference in amplitude persisted between full-reference and no-reference in-phase motion for both horizontal (Fig. 2b) and vertical conditions (Fig. 2g).

Experiment 3 used the same conditions as Experiment 1, but the endpoints of the monocular apparent motion trajectories were manipulated (see Fig. 1d). This has a strong effect on the percept and first harmonic responses produced by the anti-phase conditions (see below), but produces *2F* responses similar to those in Experiment 1, for all conditions. We again saw much weaker responses for unreferenced compared to referenced in-phase motion for both horizontal and vertical conditions (see Fig. 2c, h).

We tested the significance of the difference between full-reference and noise-reference in-phase motion with paired two-way *t* tests comparing the projected SSVEP amplitude at each displacement (see Methods), as well as the average across all displacements. This statistical approach is applied for all comparisons of SSVEP amplitudes reported here, and in most cases, we report only the displacement averages. The complete results, including effect sizes, for all displacements and averages are reported in Supplementary Tables. For Experiments 1–3, in-phase motion consistently produced larger SSVEP amplitudes with the full reference than with the noise reference for both horizontal and vertical displays (displacement averages; all *p*'s <0.0001; see Supplementary Table 1).

Like in-phase motion, anti-phase motion evoked a response that is a saturating function of displacement and noise-reference responses were strongly reduced (compare dark and light green in Fig. 2a, f). We found that for the anti-phase conditions run in Experiments 1–3, the full reference produced consistently larger responses compared to the noise reference, for both horizontal and vertical displays (weakest displacement-average effect: $p = 0.0008$; see Supplementary Table 2). These results demonstrate that our paradigm is sensitive to relative motion-specific responses and highlight the importance of a reference for both in-phase motion and anti-phase motion processing, regardless of whether the motion is horizontal or vertical.

**Horizontal and vertical anti-phase motion is suppressed**. In the horizontal conditions, full-cue anti-phase motion in the two eyes creates IOVD and CDOT cues that support a percept of MID. This is not the case for vertical anti-phase motion or for in-phase motion regardless of direction. We now focus on specifically binocular mechanisms, by comparing conditions that only differ in their interocular phase. The response functions for the full-cue/full-reference anti-phase conditions were shifted rightward on the displacement axis relative to the in-phase conditions, suggesting

that the visual system was less sensitive to anti-phase motion. The five adult experiments (1, 2, 3, 4, and 5) all had virtually identical full-reference in-phase and anti-phase conditions, so we tested for significant differences between in-phase and anti-phase responses for each of them (see Supplementary Table 3). Displacement-average amplitudes were significantly larger for in-phase motion compared to anti-phase motion for the horizontal conditions in Experiments 1–4 (weakest effect: $p = 0.0028$). For Experiment 5, the displacement average was not significantly larger for in-phase compared to anti-phase ($p = 0.2541$); however, three out of ten displacement values did produce a significant difference (strongest effect: $p = 0.0049$). For vertical conditions, all five experiments produced significantly stronger in-phase displacement averages (weakest effect: $p = 0.0444$). The differences between in-phase and anti-phase responses are thus mostly stable across the full-cue/full-reference conditions that were repeated in multiple experiments.

The no-reference conditions in Experiment 2 produced stronger responses than the noise-reference conditions used in Experiments 1 and 3, and unlike the full-reference conditions, the anti-phase response function is no longer shifted to the right of the in-phase response function, but is rather shifted to the left (Fig. 2b, g). This manifests as a stronger response to no-reference anti-phase compared to in-phase motion for both horizontal and vertical conditions (see Supplementary Table 4). For the horizontal conditions, this effect did not reach significance for the displacement average ($p = 0.1717$), but was significant at four out of ten displacements (largest effect: $p = 0.0007$). For the vertical conditions, the effect reached significance for the displacement average ($p = 0.0003$) and at five of ten bins. Taken together, these results indicate that the suppression of responses to anti-phase motion is independent of whether the displays are horizontal or vertical, but does depend on the presence of a stable reference.

**Perceptual suppression is not specific to MID**. In our measurements, full-reference anti-phase responses are reduced relative to in-phase responses for both horizontal conditions that elicit a percept of stereoscopic motion and vertical conditions that do not. This effect may be related to the perceptual phenomenon known as "stereo-movement suppression,"[42] a reduction in displacement sensitivity under binocular viewing conditions that has been replicated numerous times and is usually attributed to a stereoscopic motion system[23–26]. To connect our electro-physiological results to perception, we conducted two psychophysical motion detection experiments using the method of ascending and descending limits. In the first, participants viewed the full-cue/full-reference conditions from Experiments 1 and 2. In the second, participants viewed the full-cue/no-reference conditions from Experiment 2. To directly link perceptual data to the SSVEP response functions, we recorded SSVEPs during the psychophysical measurements. The SSVEP data from both psychophysical experiments were projected through the first reliable component generated by reliable components analysis (RCA) performed on the *2F* data from Experiment 2, and averaged over ascending trials and flipped versions of the descending trials.

In the first psychophysical experiment, the range of displacements was decreased to 0.25 to 4 arcmin to place the smallest displacements of the sweep below perceptual threshold (see Fig. 3). The results followed the same pattern seen in the previous experiments that used suprathreshold displacements (see Fig. 2): SSVEP amplitudes were larger for in-phase compared to anti-phase motion. Displacement averages were significantly larger for in-phase, for both horizontal and vertical conditions (both *p*'s <0.001; see Supplementary Table 6). Psychophysical detection

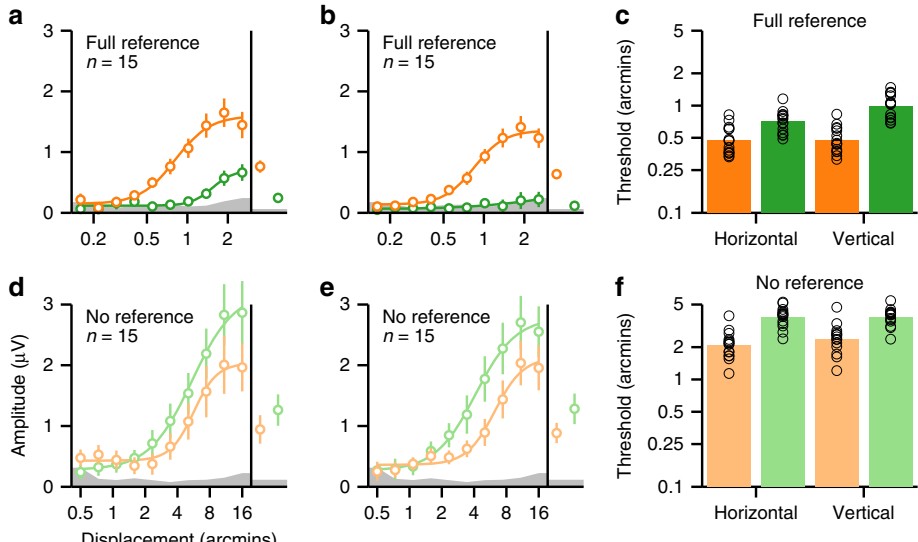

**Fig. 3** Adult second harmonic SSVEP response functions and psychophysical detection thresholds. **a**, **b** SSVEP data from horizontal and vertical full-cue/full-reference ($n = 15$). In-phase conditions are plotted in orange and anti-phase conditions in green. We ran both descending and ascending displacement sweeps; response functions are averages over ascending and flipped versions of descending trials. As in Fig. 2, averages across all displacements are shown on the right side of the response functions, gray bands represent the average background EEG noise, and smooth curves are Naka–Rushton fits to the data. Responses were weaker for anti-phase compared to in-phase for both horizontal and vertical motion. **c** Psychophysical thresholds for in-phase (orange) and anti-phase (green) conditions for horizontal and vertical directions of motion, plotted on a log scale, and again averaged over ascending and descending sweeps. Individual participant thresholds are plotted as circles overlaid on the bars. Thresholds were higher for anti-phase compared to in-phase motion. **d**, **e** SSVEP data for no-reference conditions ($n = 15$). Note that a larger range of displacements was used for the no-reference conditions and that the response functions depart from the noise level at higher displacements than in the full-reference conditions. Unlike the referenced conditions, responses are weaker for in-phase compared to anti-phase. **f** Psychophysical thresholds for the no-reference condition. Overall psychophysical thresholds are higher by a factor of ~4.7 than for the full-reference conditions and thresholds are higher for anti-phase than in-phase motion for both orientations. Error bars are on the SSVEP data are ±1 SEM

thresholds, averaged over ascending and descending sweeps, were higher for anti-phase motion by factors ~1.5 for horizontal and 2 for vertical displacements (Fig. 3c), highly significant effects in paired two-way $t$ tests (weakest effect: $p = 0.0001$), indicating that our experimental conditions do give rise to the so-called stereo-movement suppression phenomenon.

The second psychophysical experiment used the displacement range from the main experiments, under the assumption that unreferenced thresholds would be higher. This is what we observed: SSVEP thresholds across conditions were on average higher for the no-reference data by a factor of ~5.3, compared to the full-reference data from the first psychophysical experiment. The SSVEPs replicated the reversal observed in the results from Experiment 2 (compare Fig. 2b, g with Fig. 3d, e). The no-reference conditions produced significantly stronger displacement-average amplitudes to anti-phase compared to in-phase, for both horizontal and vertical conditions (weakest effect: $p = 0.0169$; see Supplementary Table 6). The psychophysical thresholds for the no-reference data, averaged over ascending and descending sweeps, were higher than for the full-reference data by a factor of ~4.7, but led to comparable differences between in-phase and anti-phase motion. Anti-phase thresholds were higher than in-phase by factors ~1.8 for horizontal and ~1.6 for vertical (Fig. 3c), highly significant effects in paired two-way $t$ tests (both $p$'s <0.0001), indicating that the perceptual stereo-movement suppression phenomenon persisted in the no-reference conditions.

The results of the first psychophysical experiment extend the pattern seen over the larger range of displacement amplitudes in Fig. 2 to the threshold regime. For both horizontal and vertical directions of motion, SSVEPS are weaker and perceptual thresholds are higher for anti-phase compared to in-phase conditions. Because the vertical conditions do not give rise to a percept of MID, we conclude that the suppressed anti-phase responses are not uniquely associated with percepts of MID. In the second psychophysical experiment, the SSVEP data replicate the results of Experiment 2, with overall weaker responses for no-reference conditions, and slightly stronger responses for anti-phase than in-phase, for both horizontal and vertical conditions. Psychophysical thresholds are higher for no-reference than full-reference conditions, but in contrast to the SSVEP data, thresholds are higher for anti-phase conditions than in-phase. This discrepancy can perhaps be explained by the presence of subtle reference cues in the experimental environment that were not encoded by most of the neurons generating the population response measured by the SSVEPs, but could still be picked up by participants. Note that vergence eye movements are unlikely to be a factor here, given that the frequency of the apparent motion (2 Hz) is too fast for vergence tracking[43]. In any event, the SSVEP data are consistent with the conclusion of Experiment 2: the suppression of responses seen with anti-phase motion depends on the content of the reference region and is independent of whether the displays are horizontal or vertical.

**Infants are insensitive to references and interocular phase**. We presented the horizontal full-cue/full-reference and full-cue/no-reference displays to 22 infants (~5 months old) and found several differences between the infant and adult data. First, infant $2F$ responses were independent of whether the reference bands contained static dots or were blank (see Fig. 4), in distinct contrast to the adult data (see Fig. 2); for both in-phase and anti-phase conditions, displacement averages were not significantly different between full and no-reference (strongest effect: $p = 0.262$; see Supplementary Table 7). Only one displacement value

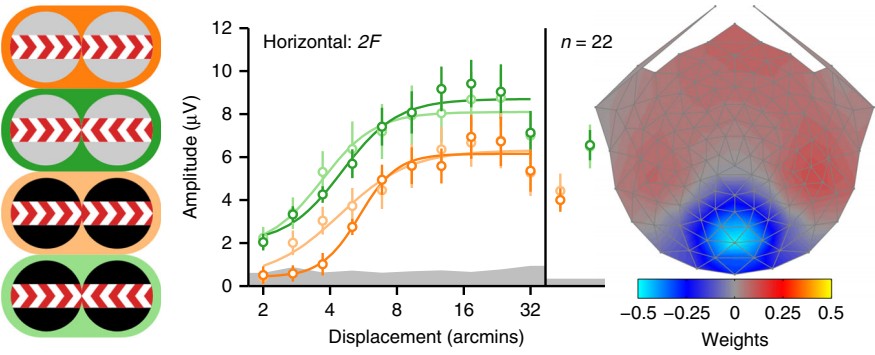

**Fig. 4** Infant second harmonic SSVEP response functions. Averages across all displacements are shown on the right side of the response functions. As for the adult analyses, data are from the first reliable component from an RC analysis derived from *2F* data from all conditions. The scalp topography for this component is shown on the right, with the color scale indicating the component weights. The conditions are represented with the icons on the left, with the same color convention as in Fig. 2. Smooth curves are fits from a Naka–Rushton function. Error bars are ±1 SEM, and the gray band indicates the average EEG noise level

reached significance, and did so in the direction opposite of what was seen for adults, indicating larger amplitudes for no-reference ($t(19) = -2.7236$, $p = 0.0135$). No displacements reached significance for anti-phase conditions (all $p$'s > 0.2). Second, infant anti-phase responses were larger than in-phase responses, in a reversal of the adult response pattern. This was true for both full-reference and no-reference conditions (both displacement average $p$'s <0.0001). Note that we observed similar reversals for adults for the no-reference conditions run in Experiment 2 (see Fig. 2b, g) and the second psychophysical experiment (see Fig. 3). The infant response pattern with both full-reference and no-reference thus resembles the adult no-reference response. Importantly, there is no evidence for suppression of referenced anti-phase responses relative to in-phase ones.

**Anti-phase suppression is a property of the IOVD system.** Perceptual stereo-movement suppression has previously been explored with stimuli that have both IOVD and CDOT cues[23–25,32,42]. For the full-cue conditions just described, IOVD and CDOT are present in both horizontally and vertically oriented displays, but only the horizontal versions of the two cues can support a computation of MID[22]. Following this asymmetry, the literature on CDOT and IOVD has focused on the horizontal case, but the neural signature of anti-phase suppression we see in our data can be measured for both orientations. Because IOVD is an explicitly motion-based cue, we wanted to determine whether IOVD cues alone could elicit suppression.

We isolated the IOVD cue using two previously developed approaches. In Experiment 4, we used dots in the moving regions that were uncorrelated between the two eyes (IOVD-u)[44–46]. In Experiment 5, we used dots that were anti-correlated between the two eyes (IOVD-a)[47]. Each manipulation defeats the binocular matching necessary for extracting CDOT. Although the suppression effects were weaker and less consistent for the IOVD-isolating conditions, we nonetheless saw evidence that anti-phase suppression can be generated by the IOVD cue alone (see Supplementary Table 5). For horizontal and vertical IOVD-u (purple and blue in Fig. 2d, i), the displacement averages were stronger for anti-phase compared to in-phase (weakest effect: $p = 0.0039$). We also observed significant anti-phase suppression for IOVD-a, for both horizontal and vertical displacement averages (weakest effect: $p = 0.0214$).

Diminished suppression effects may occur because the uncorrelated and anti-correlated conditions cause a different mixture of disparity-related and motion-related responses. The

anti-correlated cue is expected to activate disparity-tuned cells in early visual cortex, with an inverted sign[11]. This would not be expected for the uncorrelated case. The two conditions may thus cause a different mixture of disparity-related and motion-related responses. Moreover, both IOVD-isolating conditions can trigger binocular rivalry that may reduce the response magnitude in the in-phase condition that is used as the reference for the suppression effect (compare orange and purple in Fig. 2d, e, i, j). Nonetheless, the overall effects and trends in the data indicate that anti-phase suppression can be generated by the IOVD cue alone.

**4F response is a candidate signal related to MID from IOVD.** The evoked response contains multiple even and odd-harmonic response components. We have focused on *2F*, which behaved similarly for horizontal and vertical directions of motion and is thus not a likely source of MID signals for perception. To look further for a possible marker for evoked responses that could contribute to MID from IOVD, we examined both the *2F* and *4F* SSVEPs for evidence of differential responses to horizontal and vertical directions of anti-phase motion, under the assumption that components contributing to MID should exhibit differential responses to perceptually relevant and irrelevant directions of motion. We combined the data from the full-cue/full-reference horizontal and vertical anti-phase conditions across Experiments 1, 2, 3, and 4, yielding a data set with 42 participants (note that for individuals that took part in several experiments, only their first session was added to the combined data set). We then derived reliable components over the larger group separately for the *2F* and *4F* data, using the same RCA approach as for the individual experiments. Figure 5 shows *2F* (a) and *4F* (b) response functions for the two orientations of anti-phase motion, from the first reliable component. The horizontal response functions are plotted in green, with the vertical data in red. The component topographies are plotted in (c) and (d). As expected, the *2F* topography is very similar to the *2F* topographies generated separately for each experiment (compare Supplementary Fig. 2 and Fig. 5c), while the *4F* topography was perhaps slightly broader (Fig. 5d). There was no measurable difference between responses to horizontal and vertical orientations at *2F*, whereas for *4F* the green curve is shifted to the left, indicating that for the perceptually relevant horizontal stimulus orientation, responses rise out of the noise at lower displacements. This *4F* effect manifested as larger responses to horizontal motion at five of ten displacements (strongest effect: $p = 0.0034$), but no significant

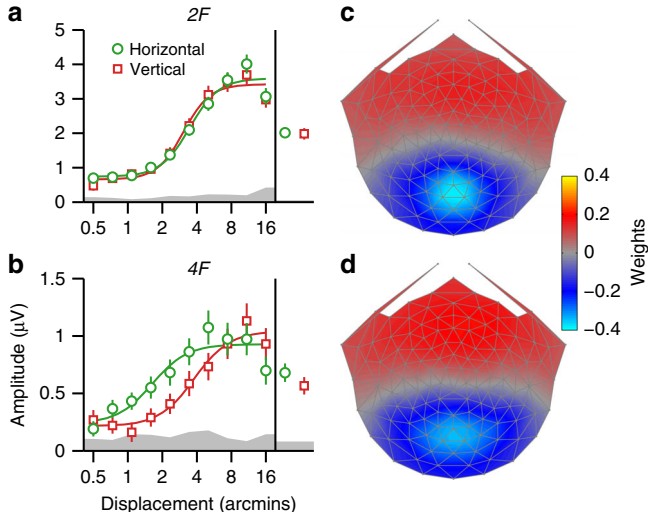

**Fig. 5** Candidate MID signal from IOVD. Response functions for horizontal (green) and vertical (red) full-cue/full-reference anti-phase motion conditions, averaged across Experiments 1, 2, 3, and 4. Averages across all displacements are shown on the right side of the response functions. The response functions are from the first reliable component of RCA done separately on 2F (**a**) and 4F (**b**) data, with the topographies shown on the right (**c**, **d**). The color scale indicates the component weights. Responses do not differ for the two directions of motion at 2F, but are different at 4F at the smaller displacements. An analogous analysis done on the uncorrelated and anti-correlated anti-phase conditions from Experiments 4 and 5 produced a similar pattern of results (see Supplementary Fig. 3). Error bars are ±1 SEM, smooth curves are Naka–Rushton fits and the gray band indicates the average EEG noise level

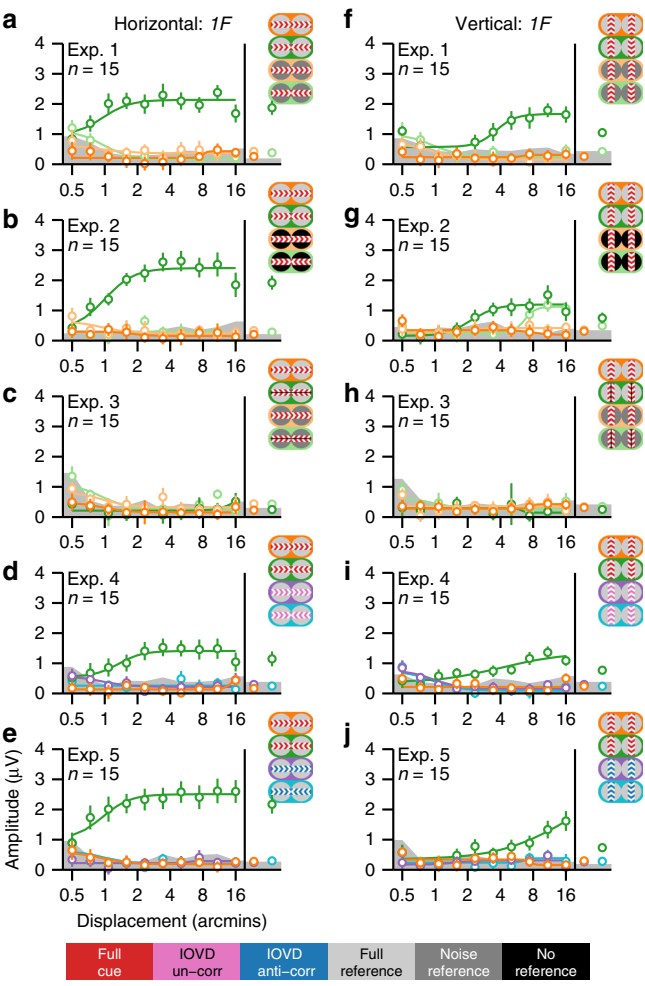

**Fig. 6** Adult first harmonic SSVEP response functions. **a–e** Displacement response functions for the horizontal direction of motion and **f–j** the vertical direction of motion. Averages across all displacements are shown on the right side of the response functions. Data are from the first reliable component from an RC analysis derived from 1F data from all conditions, separately for each experiment. Scalp topographies for this component are shown in Supplementary Fig. 2. Each experiment had eight conditions, of which half were horizontal and the rest were vertical. The conditions are represented with icons, with the same color convention as in Fig. 2 and the color meanings shown in the legend in the bottom of the figure. Smooth curves are Naka–Rushton function fits to the data and gray bands at the bottom of the plots indicate the average background EEG noise level. Error bars plot ±1 standard error of the mean (SEM). As expected, Experiment 3 produced no reliable 1F response for any condition, see text for additional details

effect for the average (see Supplementary Table 8). For 2F, one displacement reached significance ($p = 0.0276$) and another was close ($p = 0.0553$), but all other comparisons were far from significance (all $p$'s >0.15).

If the 4F effect is in fact a substrate for MID from IOVD, we would expect it to replicate in the IOVD-isolating conditions. To test this, we repeated the analysis with combined data from the IOVD-u and IOVD-a conditions run in Experiments 4 and 5, yielding a data set with 24 unique participants. The first reliable components for 2F and 4F were similar to those produced for the larger data set (see Supplementary Fig. 3C, D), and the overall trend in the SSVEP response functions was similar (Supplementary Fig. 3A, B). There was a significant difference between horizontal and vertical motion for 4F displacement averages ($p = 0.0053$; see Supplementary Table 8). The 2F average response was far from significance ($p = 0.4808$), and the only significant effect was in the opposite direction, in favor of vertical.

Overall, these results are consistent with the hypothesis that neurons generating the 4F response can support a percept of MID. This might occur if the MID system involves a sequential cascade of non-linear operations. The relevance of 4F to MID perception needs to be tested by experiments examining co-variation of the 4F response with depth percepts.

**Image segmentation responses are driven by CDOT.** The anti-phase condition is phenomenologically asymmetric—for horizontal motion, the observers' percept alternates between segmented planes of disparate bands and a flat plane (zero disparity over the whole image). As mentioned above, this asymmetry in perceptual organization can manifest as an evoked response at the first harmonic (1F) of the stimulus frequency, that is, the rate at

which the perceptual organization changes (2 Hz). The 1F displacement response functions are shown in Fig. 6 for the five main experiments. The data are from the first reliable component produced by RCA performed separately on the 1F data, using the same approach as for 2F.

Like the 2F responses, the 1F response is a saturating function of displacement, starting from the smallest displacement amplitude that depends on the presence of a local reference of static dots. In the horizontal full-cue/full-reference conditions of Experiments 1, 2, 4, and 5 there is strong 1F response for anti-phase motion, (Fig. 6a, b, d, e, dark green). This response is a relative disparity response as it is obliterated for both noise reference and no-reference (light green), and for the in-phase conditions where there is no MID (dark and light orange). Similar dependence on a correlated zero

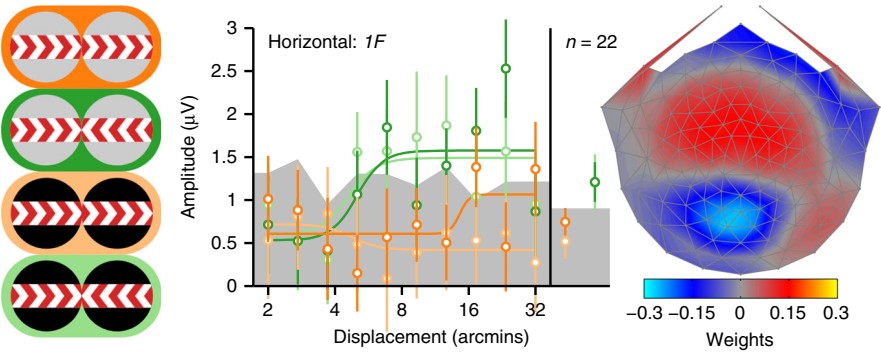

**Fig. 7** Infant first harmonic SSVEP response functions. Averages across all displacements are shown on the right side of the response functions. Unlike previous plots, data are plotted for the fifth reliable component because its topography was most similar to the topography of the adults. The scalp topography for this component is shown on the right, with the color scale indicating the component weights. The conditions are represented with the icons on the left, with the same color convention as in previous figures. Smooth curves are fits from a Naka–Rushton function. Error bars are ±1 SEM, and the gray band indicates the average EEG noise level

disparity reference has been observed with dynamic random-dot displays that fully isolate the CDOT cue[35].

We also obtained measurable *1F* responses for vertical conditions with a static dot reference, but the response function is shifted to the right by a factor of ~4 (Fig. 6f, g, i, j, dark green). Here the stimulus contains vertical relative disparity. The weak and absent *1F* responses in the noise and no-reference conditions (Fig. 6f, g, light green), indicate that this is indeed a relative disparity response. A vertical relative disparity signal has also been found with pure CDOT dynamic random-dot stimuli[35]. In both cases, displacement sensitivity is about four times better for horizontal than for vertical disparity. The importance of references is well known for horizontal disparity[48–50]. The present results suggest that relative disparity is also calculated along the vertical direction, consistent with our previous results[35], and provide support for a previous psychophysical finding that vertical disparities can be used for discontinuity detection[51].

If *1F* is in fact due to the phenomenological asymmetry over time as the anti-phase display alternates between uniform and segmented percepts, the *1F* response should disappear with a stimulus configuration in which the display no longer alternates between uniform and segmented states. Such a configuration also tests the alternative hypothesis that the asymmetry leading to *1F* is due to different response amplitudes for motion towards and away from the observer. For Experiment 3, we generated such a display by making the two endpoints of the monocular apparent motion trajectories straddle zero disparity, which means that the bands alternated between equal and opposite values of crossed and uncrossed disparity. This subtle manipulation eliminates the asymmetry associated with alternating between uniform and segmented percepts, but still gives rise to motion towards and away from the observer in the horizontal anti-phase conditions, and to left/right or up/down motion in the plane in the in-phase conditions. The latter point explains why *2F* responses are so similar between Experiments 1 and 3. The *1F* response is eliminated in Experiment 3, for both horizontal and vertical directions of motion, tying the response to asymmetric processing of the uniform vs. segmented stimulus states (Fig. 6c, h).

In the full-cue condition, the *1F* response could in principle arise from either the CDOT or IOVD cue, as both are present. However, the *1F* response becomes unmeasurable in the IOVD-u (Fig. 6d, i) and IOVD-a (Fig. 6e, j) conditions that eliminate the CDOT cue, indicating that *1F* responses are driven by CDOT. The fact that this CDOT-driven *1F* response can be measured for vertical relative disparities suggests that it is not exclusively a MID signal, at least at large disparity values.

Finally, in the case of the infants, a weak, but measurable *1F* response was present in the full-cue/full-reference horizontal condition (Fig. 7, dark green). The sensitivity to displacement in our full-cue display here is similar to our previous measurements with CDOT-isolating dynamic random-dot stereograms[35]. In contrast to the adult data, the infant *1F* response was also measurable in the no-reference condition (Fig. 7, light green), although caution is needed here given the weak responses overall.

## Discussion

Our data provide new insights into early motion processing stages. For in-phase conditions that give rise to motion in the fronto-parallel plane, we show that both threshold and supra-threshold responsivity is strongly dependent on the presence of a nearby reference in adults, but not in 5-month-old infants. Infants were not sensitive to relative motion under our stimulus conditions. The pattern of activity we observe is consistent with relative motion being processed via directionally opponent interactions between the classical receptive field and its non-classical surround[5–7,52]. The lack of reference effects we observe in infants may reflect an immaturity in these interactions.

For anti-phase motion, we identify an IOVD-based mechanism that operates on both horizontal and vertical motion directions. The functional manifestations of this mechanism are an elevation of perceptual threshold for anti-phase motion relative to in-phase motion and a decrease in SSVEP response amplitude. Amplitude reductions persisted with IOVD-isolating stimuli, linking the phenomenon to that cue. Because suppression is equal for horizontal and vertical directions of motion, it is unlikely to be related to the extraction of MID, per se, because stimulus information for MID from IOVD is only present for horizontal or near-horizontal directions of motion[22]. Prior descriptions of the suppression phenomenon as reflecting stereoscopic depth movement[23] thus need to be revised. The observed suppression either precedes the computation of MID from this cue or operates in parallel.

The anti-phase suppression we have observed is consistent with a dichoptic, directionally opponent interaction. The functional form of the suppressive interaction is a rightward shift of the response curve on the displacement axis, consistent with a form of divisive normalization[53]. Prior work has suggested that the perceptual stereo-movement suppression effect is due to limits in sensitivity imposed by increased noise in binocular differencing operations[24,25,32]. By using a direct neural measure over both threshold and suprathreshold levels, we see that the stimulus-

driven population response itself is strongly attenuated. This attenuation is difficult to accommodate within a probabilistic, noise-limited detection framework. Suppression is more consistent with mutual inhibition between the two eyes, as initially suggested in the first observations of the suppression effect[23].

Motion responses in V1 have traditionally been modeled with variants of an energy-like metric[54–56] that were not designed to explain human observers' greatly enhanced sensitivity to relative motion[8–10]. This limitation can be addressed by looking towards an existing model of relative disparity processing, which incorporates opponent-based pooling of V1 disparity energy units and is successful in explaining several properties of V2 cell responses to disparity edges[57]. Formulating a relative motion model along these lines, with motion energy substituted for disparity energy at the first stage, would certainly be feasible and thus the effects of references in both motion and disparity domains could be accommodated by an analogous two-stage model. Existing physiological and functional magnetic resonance imaging (fMRI) data suggest that both stages are present in V1 for motion[58–60], but that the second stage of the relative disparity system begins in V2 or beyond[17,60–62]. The present results suggest that for both motion and disparity (in line with our previous results[35]), this hypothesized second stage is immature in infants. In the case of motion, non-classical surround effects[5–7,52,58] are viable candidates for an underlying mechanism for enhanced responses to relative motion. Analogous non-classical surround effects in the disparity domain have been observed in macaque middle temporal[63] and medial superior temporal[64] areas. Human fMRI data also provides evidence for opponency in the disparity domain, starting no later than V3[60].

Disparity tuning in macaque V1 is distributed over all orientations and is thus not specifically associated with horizontal disparities that support stereopsis[65,66]. Relative disparity tuning in single cells, to our knowledge, has only been probed with horizontal disparities[16–19,67]. The present results suggest that sensitivity to relative vertical disparity is present at least for larger disparities presented over a relatively large field-of-view (up to 20° eccentricity). Our neural correlate of relative disparity processing is the first harmonic response (1F). We showed 1F arises from CDOT rather than an IOVD because the response is eliminated in the two stimulus conditions (interocularly uncorrelated and anti-correlated dots) that remove percepts of MID from CDOT, but not from IOVD. Functionally, our candidate vertical relative disparity signal is robust when a reference is present and is eliminated when the reference is removed, as is the case for horizontal disparity. Further corroboration comes from Experiment 3 where disparity modulates symmetrically around the reference. This manipulation specifically varies the relative disparity between the reference and moving bands as well as the global stimulus configuration. Uniform and segmented states could thus be differentiated based on the 1F signal for both horizontal and vertical stimulus orientations. A second-stage pooling of vertical absolute disparities could give rise to vertical relative disparity sensitivity in the same way as has been suggested for horizontal relative disparity[57]. Computationally, this model implements a form of opponency in the disparity domain, but the opponency is within the classical receptive field of second-stage V2 neurons, rather than between classical receptive field and its surround. Whether vertical relative disparity sensitivity is a property of the classical receptive field or center–surround interactions remains to be determined. A characteristic behavior that any model of relative disparity processing will need to account for is the fact that sensitivity is dramatically greater for horizontal compared to vertical relative disparity by a factor of at least 4 in the adults (Fig. 6). Whether the difference is due to the properties of first-order cells in V1 or whether it represents a

specific adaptation to horizontal disparities in higher visual areas remains to be determined.

Different models of MID have compared architectures where disparity is extracted first, followed by a second stage that computes CDOT with architectures where velocity is computed first and then differenced at a second stage (IOVD)[68,69]. We have presented evidence for each of these processes. The existing models were conceived in the context of MID from horizontal disparity/motion. Our IOVD correlate, the suppression of the second harmonic (2F) response for anti-phase motion, is present for both horizontal and vertical displays. The same is true for our CDOT correlate in 1F. Thus, neither of these visual responses are specific to horizontal motion or a percept of MID. This suggests that CDOT and IOVD cues generate robust responses that are insufficient to produce a percept of MID. Rather, we have identified what appears to be an intermediate set of signal processing operations that either occur before MID extraction or operate in parallel with it. Moreover, none of the existing models of CDOT or IOVD can account for the influence of references that we demonstrate here. The suppression we observe here may be related to a previously described "fusional suppression" phenomenon that also operates for both horizontal and vertical displays[70] and also involves suppression of monocular information in a dichoptic presentation. Another possibly related finding comes from work on vergence eye movements indicating that the IOVD cue can elicit both horizontal and vertical vergence[71], suggesting another role for IOVD besides perception of MID.

Which responses in our data are likely to reflect activity specifically related to MID? Evidence for an MID signal based on CDOT comes from our finding that disparity thresholds for the CDOT-specific 1F response are much lower for horizontal than for vertical conditions, consistent with the prominent role that horizontal relative disparity plays in perception. We see very little evidence for an asymmetry between horizontal and vertical conditions at 2F, our IOVD correlate. We do, however see a measurable superiority of responsiveness to smaller horizontal displacements at the fourth harmonic (4F) that persists for stimuli that isolate the IOVD cue. The fact that 4F—but not 2F—is tuned for stimulus orientation indicates that the two harmonics are not being generated by a common higher-order non-linearity, such as rectification in a population of transient or direction-selective cells. An alternative view that is qualitatively consistent with velocity-first MID models is a cascade model in which a first-stage motion computation creates 2F responses at its output that are then pooled in a non-linear fashion by a binocular MID process, with the result being a response that is fourth order with respect to the input. In this model, the 2F response would have to be monocular for the fourth harmonic to represent the output of the binocular MID stage. However, 2F is modulated by interocular phase and is thus at least partly due to binocular mechanisms. This suggests that MID and anti-phase suppression may arise in parallel pathways, rather than being properties of a common MID mechanism.

The visually evoked responses of 5-month-old infants in the presence of high-quality motion and disparity references bear a strong resemblance to the unreferenced responses of adults, suggesting that both relative motion and relative disparity mechanisms are selectively immature in infants. In addition to their lack of sensitivity to references, infants also show a lack of anti-phase suppression. These are qualitative immaturities that cannot be explained by lower overall displacement sensitivity in infants. Because anti-phase suppression in adults is strongly dependent on the presence of references, it is conceivable that the lack of anti-phase suppression in infants is driven by their lack of sensitivity to references. However, it is also possible that the lack of anti-phase suppression represents a separate immaturity

relating to binocular vision. In-phase motion responses were measured under binocular conditions, but would likely be very similar if presented under monocular viewing conditions.

## Methods

**Participants and procedure**. A total of 59 adults participated in one or more of the experiments (age range 17.1–40.8 years; mean 23.2, SD = 5.24). Twenty-two healthy full-term infants with birthweights exceeding 2500 g participated (12 male, avg. age = 5.6 months, SD = 1.1). The adults had normal or corrected-to-normal vision, with a visual acuity of 0.1 LogMar in each eye or better and no significant difference of performance between both eyes. Adult participants also scored at least 40 arcsec or better on the RanDot Stereogram test. Prior to the experiment, the procedure was explained to each participant or the parent, and written informed consent was obtained before the experiment began. The protocol was approved by the Stanford University Institutional Review Board. Adult participants were solicited though Stanford University subject pools. Infants were recruited via mailers sent to addresses procured by the California Department of Public Health, Center for Health Statistics and Informatics.

**Stimuli**. In all experiments, participants viewed random-dot kinematograms or stereograms on a 65-inch Sony Bravia XBR-65HX929 LCD monitor. The dots were drawn with OpenGL using anti-aliasing at a screen resolution of 1920 × 1080 pixels. This function allowed us to present disparities via dithering that were smaller than the nominal resolution set by the 1920 × 1080 display matrix. This was verified by examining the contents of video memory and through examination of the anti-aliased pixels under magnification. The dots were updated at 20 Hz.

For seven of eight experiments, viewing distance was set at 1 m, resulting in a 40° × 40° field of view, a 4.5 arcmin dot diameter, with 5 dots per square degree. In the first psychophysics experiment, the viewing distance was set at 3 m, to allow for smaller increments in dot locations. The stimuli were rendered as Red/Blue anaglyphs. The luminance of the images in the two eyes was equated by calibrating the display through each filter. A −0.5 D lens was placed in the Blue channel of the adult glasses to compensate for the differential focus caused by chromatic aberration. Cross-talk was minimal perceptually when viewing high-contrast images in the two eyes.

**Experimental design and procedure**. Schematic illustrations of the stimuli are provided in Fig. 1. The displays in each experiment consisted of alternating bands containing random dots that could differ in their interocular correlation, temporal correlation, or both. One set of "test bands" underwent in-phase or anti-phase motion on each trial, with the adjacent "reference bands" serving as a static or dynamic reference. There were 20 test bands and 20 reference bands on the screen (spatial frequency of 0.5 c/deg). When the interocular correlation was 100%, there were matching dots in each eye, when it was 0%, independently generated dots were presented to each eye. When the interocular correlation was −100%, the dark dots in one eye were matched with bright dots in the other eye (see Supplementary Methods for further details). Displays in which the temporal correlation was 100% had dots that moved coherently with unlimited lifetime. Displays that had a temporal coherence of 0% had newly generated dots on every image update (20 Hz). Dots were replotted at the end of motion trajectory to keep the number of dots on the screen the same at all times and the location of the borders of the dot region constant.

Experiments 1, 2, and 3 used a 2 × 2 × 2 design, with factors of interocular phase, reference quality, and stimulus orientation, resulting in eight conditions. Interocular phase of the moving test bands was either in-phase or in anti-phase, that is, either a 0° or 180° temporal phase relationship applied between eyes. Manipulation of the dots in the reference region created full-reference and unreferenced motion and disparity conditions. In Experiments 4 and 5, manipulation of the interocular correlation in the moving bands (changing them from 100% correlated between eyes to either 0% or −100%) was used to isolate the IOVD cue. Here the design was also 2 × 2 × 2 with factors of interocular phase, interocular correlation, and orientation. In each of the adult experiments, participants completed 15 trials per stimulus condition, and the eight conditions were run in a block-randomized fashion: a block consisted of five consecutive trials of a given condition. Data were collected in three separate, continuous recording sessions, each lasting approximately 8 min, with a break given in between. During each session, a single block of each condition was run. Observers were instructed to fixate the center of the display and to withhold blinks. For the infant experiment, only the four horizontal conditions were used, resulting in a 2 × 2 design, with factors of interocular phase and reference quality. Infants completed 10 trials. Details of each experiment are provided in the Supplementary Methods.

**EEG acquisition and processing**. Data were collected from all participants using high-density HydroCell electrode arrays paired with an Electrical Geodesics NetAmp400 and accompanying the NetStation 5 software. The nets used for adults had 128 channels and the one used for infants had 124. Electroencephalograph (EEG) data initially sampled at 500 Hz was resampled at 420 Hz to provide seven samples per video frame. Digital triggers were sent from in-house stimulus

presentation software and stored with the EEG recording to allow synching of the visual stimulus and EEG with millisecond precision. Recordings were exported from NetStation using a 0.3–50 Hz bandpass filter, which was applied twice to ensure that power in frequencies outside the filter range was minimized. The data were then imported into in-house signal processing software for preprocessing. If more than 15% of samples for a given sensor exceeded an amplitude threshold, the sensor was excluded from further analysis. Adult data were evaluated using a 30 μV threshold, whereas a more liberal threshold was applied for the infant data, ranging between 30 and 100 μV. Sensors that were noisier than the threshold were replaced by an averaged value from six of their nearest neighbors. The EEG data were then re-referenced to the common average of all the sensors and segmented into ten 1000-ms-long epochs (each corresponding to exactly 2 stimulus cycles). Epochs for which more than 10% of data samples exceeded a noise threshold (30 μV for both adult and infant participants) or for which any sample exceeded a peak/blink threshold (60 μV for both adult and infant participants) were excluded from the analysis on a sensor-by-sensor basis. If more than seven sensors had epochs that exceeded the peak/blink threshold, the entire epoch was rejected for all channels. This was typically the case for epochs containing artifacts, such as blinks or eye movements.

**Data analysis**. In our sweep paradigm, stimulus values were updated for every 1 s bin, so each epoch in our analysis is tied to a distinct set of stimulus parameters, for a given trial. The amplitude and phase of the SSVEPs were extracted using a recursive least-squares adaptive filter[72] with a memory length equal to the 1 s bin length. Real and imaginary components of the SSVEPs at the first four harmonics of the stimulus frequency were calculated. Noise estimates were calculated at neighboring frequency bins, that is, $F ± 1$ Hz.

We reduced the spatial dimensionality of our data by decomposing the sensor data into a set of physiologically interpretable components using reliable components analysis (RCA)[73]. Because SSVEP response phase is constant over repeated trials of the same stimulus, RCA utilizes a cross-trial covariance matrix to decompose the 128-channel montage into a smaller number of components that maximize trial-to-trial consistency through solving for a generalized eigenvalue problem. The real and imaginary values for each 1 s epoch, across the 128 sensors, and across trials and participants, served as the input data for RCA. Reliable components were derived separately for each harmonic in each of the five main experiments, and separately for each harmonic in the infant experiment. The data reduction for the SSVEP data collected during the psychophysical experiments was done by projecting the data through the component weights from the first RC derived for 2F in Experiment 2, which used the same stimulus parameters as the psychophysical experiments.

We ran two additional reliable components analyses combining data from unique participants across several experiments. One exclusively with data from the static reference horizontal and vertical anti-phase conditions from Experiments 1, 2, 3 and 4, yielding a data set with 42 unique participants, and another with data from the horizontal and vertical anti-phase conditions in the uncorrelated and anti-correlated IOVD-isolating Experiments 4 and 5, yielding a data set with 24 unique participants. In both cases, we derived reliable components separately for the 2F and 4F data, using the same RCA approach as for the individual experiments.

Our analyses focused on the first RC component, which for 2F data explained much of the reliability in the five main experiments (average = 67.2%, SD = 3.3) and a substantial amount of the variance (average = 23.0% SD = 1.8). This was also the case for 1F in four out of five experiments, excluding Experiment 3 where 1F were not measurable (average reliability explained = 50.6% SD = 4.0; average variance explained = 16.0% SD = 3.2). For the infant 1F data, the first RC did not look like a visual response, likely because SSVEPs were weak overall. We did however see a topography and response function that resembled that observed for adults in the fifth RC, which we present in Fig. 7.

After projecting the epoch-level data through the RCA component weights, averages across trials in each condition and across participants were generated using vector averaging, in which the real and imaginary coefficients for a given harmonic are averaged across participants, and the amplitude is computed from the result. The vector averages were computed separately for each of the 10 bins in the displacement sweeps. Noise estimates based on neighboring frequency bins did not contribute to RCA but were projected through the component weights to allow comparison with the SSVEP data, and then averaged across trials, participants, and conditions, for each displacement. Averages across displacements were computed for both signal and noise by averaging real and imaginary components across bins, for each participant, and then vector averaging.

For illustration purposes, we fit the vector-averaged response functions with a Naka–Rushton function[41], as described in Eq. (1):

$$R = R_{max}\left(\frac{d^n}{d^n + d_{50}^n}\right) + b, \qquad (1)$$

Where R is the response, d is the displacement, and b is the baseline. $R_{max}$ (maximal response), n (exponent of the power function, >0), and b and $d_{50}$ (displacement at half $R_{max}$) are free parameters.

For statistical analysis, we determined the magnitude of the projection of each participant's vector onto the group vector average[74]. The magnitudes of these

projections were then used to compute the standard error for each condition, and to conduct paired *t* tests comparing conditions. This was done both for the individual displacement bins and the average across bins. The mean of these projected amplitudes is the same as the amplitude of vector average. The projection procedure is useful because it preserves the robust phase consistency across subjects with associated signal-to-noise ratio improvements that would not occur if amplitude means and errors and corresponding *t* tests were simply computed from individual participant amplitudes.Data availabilityThe datasets generated and analyzed for the current study, and the custom MATLAB code used for analysis, are available from the corresponding author on reasonable request.

## Data availability

The datasets generated and analyzed for the current study, and the custom MATLAB code used for analysis, are available from the corresponding author on reasonable request.

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

## Acknowledgements

This research was supported by National Institutes of Health grant EY018775. We would like to acknowledge Alexandra Yakovleva and Vladimir Vildavski for their contributions to the development of the stimulus paradigms used here. We would also like to acknowledge Marco Del Giudice, University of New Mexico, for his helpful advice on statistical approaches.

## Author contributions

P.J.K., W.J.M., and A.M.N. designed research; W.J.M. and P.J.K. performed research and analyzed data; P.J.K., W.J.M., and A.M.N. wrote the paper.Data availabilityThe datasets generated and analyzed for the current study, and the custom MATLAB code used for analysis, are available from the corresponding author on reasonable request.

## Additional information

**Competing interests:** The authors declare no competing interests.

