## [Peer Review File · Nature Communications]

Reviewers' comments:

Reviewer #1 (Remarks to the Author):

Summary:

Kohler, Meredith & Norcia used Steady-State Visual Evoked Potentials (SSVEPS), to making direct neural measurements of the divergent responses to different forms of suprathreshold motion in human adults and 5-month-old infants. They report that:

- 1) Inter-ocular differences in retinal image motion generated suppressed response functions and correspondingly elevated perceptual thresholds, compared to motion matched between the two eyes.
- 2) This suppression was of equal strength for horizontal and vertical motion and therefore not specific to the perception of motion-in-depth.
- 3) Suppression is strongly dependent on the presence of spatial references in the image, and
- 4) Suppression is very immature in infants.

They conclude "suppression appears to be the manifestation of a succession of spatial and inter-ocular opponency operations that occur at an intermediate processing stage either before or in parallel with the extraction of motion-in-depth."

Evaluation:

This paper represents a heroic set of 8 experiments; 5 SSVEP experiments and 2 psychophysical experiments in adults (N = 59), as well as SSVEP experiments in infants (N = 22). The experimental methods and analyses are sensitive and sophisticated, and the results are convincing, interesting and important to our understanding of motion in depth and perceptual suppression.

That said, I suspect that said, the multiplicity of experimental conditions (both targets and references), SSVEP components analyzed (F, 2F, 4F) and different cues to motion in depth (MID) may be confusing to non-specialized readers – indeed, I found them difficult to keep track of, and found myself having to re-read sections and going back to figure 1. One suggestion that may help the reader is to add a column to Figures 2 (and maybe Figs 3 and 4 also) replicating the bottom panel of Fig. 1. A legend in the figure for the different colors in the each panel would also be helpful, rather than having to refer to the long figure caption to determine what conditions each curve refers to. As an aside, the panels showing scalp topography get a lot of space, but almost no discussion. I wonder whether they might be better placed in Supplemental materials.

I applaud the authors for not 'slicing the salami' – and this kind of detailed analysis will be of great interest to a specialized vision science audience. However, for a general audience, it may be more helpful to strip the paper down to simply focus on the most critical conditions that highlight the main conclusions.

Some specific Issues:

Introduction: "Here we systematically explore the importance of clearly defined references on neural responses to motion and disparity using minimally complex scene structures comprised of regions defined by motion, disparity or both."

While the current work goes well beyond, it seems reasonable to reference previous work on this and closely related topics (e.g. Zemon & Vance, PNAS, 1982; Steinman et al., Vision Research, 1985; Visual Neuroscience 1992).

Results: Page 9 of the review PDF (actual page numbers would be helpful) – the Paragraph beginning "In the first psychophysical experiment . . ." "(see Figure 4)"

should be "(see Figure 3)". The next sentence asks the reader to "(compare dark green and orange in Figure 2 with Figure 4A, B)". That's hard to do since they're on separate pages and there are a lot of curves in Fig. 2 - why not just give the amplitude differences in this expt (Fig 3 - NOT 4) vs those in the previous (Fig 2).

Figure 3 and 4 – consider adding a column replicating the bottom panel of Fig. 1 showing the various conditions.

Reviewer #2 (Remarks to the Author):

The present study uses steady-state visual evoked potentials (SSVEPs) to measure the neural correlates of processing motion in depth. The main finding is that inter-ocular differences in retinal image motion resulted in attenuated neural response functions (as indexed by SSVEP amplitudes) that were accompanied by elevated perceptual thresholds, relative to motion that was matched between the two eyes.

This study entails a rich data set - electrophysiological and behavioral measurements across 8 experiments, including an infant study, and a thorough analysis of the data. However, many of the measurements used are not clearly motivated and it remained elusive to me why certain measurements of the SSVEP signal were taken into account for one interpretation or another. Furthermore, many of the statistical results ride along the p-curve - many of the p values reported are at or nearby 0.05, suggesting that the results are to be taken with caution, and the study was overall underpowered. Finally, the infant study seemed like it was simply attached to the adult study - again with no clear theoretical background or motivation. Thus, in a revision the authors should make sure to 1) induce a larger sample to power their statistical analysis appropriately, 2) provide clear theoretical background in what their measurements indicate and why they used them, 3) clarify the motivation for the additional infant experiment. Below please find more detailed points.

1) Before describing the results in detail, it would be useful to go over the experimental design and general aspects of the SSVEP measurements. In particular, why would SSVEP measurements in this case be a reasonable neural correlate of motion processing? What does it tell us beyond the behavioral thresholds? Second, why did the authors focus on second harmonics instead of first, and why did they later on in some of their analysis include 4F as well? How can we interpret different harmonics of the SSVEP? The authors often use this approach in their papers, but it would be good to clearly state what it means in this particular experiment if the responses of different harmonics differ, and how one can interpret them (ideally, a priori and hypothesis-driven, not post-hoc).

2) Figure 1 displays the different conditions but it was quite difficult to parse. Is there a way to simplify the figure? Also, I would like to see one example of an actual stimulus display used (not just the schematic illustration). In particular for the measurement of SSVEPs it's informative for the reader to see the exact stimulus that's driving the SSVEP responses.

3) Please explain why the Naka-Rusthon function was used, and whether other functions were tested.

4) It would be useful to add a few sentences about general experimental approach and measurements in the introduction to facilitate following the Results section.

5) Many, many of the statistical tests show ambiguous results: they are just significant, or approach significant, or just trending towards significance. That makes many of the results really difficult to interpret. This raises two questions: How was the number of subjects chosen (N=11 - seems very small to begin with!!) ? It looks as if this study is largely underpowered and should definitely be replicated with a larger set of participants. Second, what are the effect sizes across conditions?

6) When I followed the experimental set-up correctly it seems the stimulus input is not matched across the conditions that are later compared in the SSVEP signal. Some stimuli move, some don't. These differences will have large effects on the SSVEP amplitudes, just because of the driving stimulus, so independent of what the authors are interested in. If this is the case, differences in amplitude are difficult (if not impossible) to be interpreted as differences in perception / cognitive functions. Please clarify how this was dealt with.

7) It is unclear to me why the study on infants was included here. No (or very little) theoretical background is provided why infants could help understanding the processes of motion perception. At the same time, this study is not framed as a developmental paper. Thus, as the paper currently is written the infant data seems almost arbitrarily attached as yet another experiments without really providing new insights for the main question of interest.

8) Methods:

- Applying the same filter twice does not guarantee that that power frequencies outside the filter range are eliminated. Different filters reduce difference frequencies, but the cut-off is never perfect
- also not when applying the same filter twice. Further, since SSVEPs are the main focus of the analysis, it seems filtering is not necessary at all.

Reviewer #3 (Remarks to the Author):

I have read manuscript #.

The authors conducted a series of VEP experiments to address a confusing state of affairs which is that, on the one hand, opposite horizontal motion in the two eyes creates a percept of motion through depth (obviously important motion to be able to detect) while, on the other hand, opposing motion in the two eyes also results a reduction for motion sensitivity. The latter has always seemed completely at odds with the necessity of the former.

The authors manipulated two main things in the stimulus: the degree to which disparity cues should have been stimulated (normal moving dots vs. dots that were either un- or anti-correlated across the eyeballs) and the degree to which the moving portions of the stimulus had a reference (static, fixed disparity reference vs. either dynamic uncorrelated noise or no reference).

In addition to the main VEP experiments done in adults, VEP data were collected in 5-month-old infants, and psychophysical data were collected in adults.

Taken together, the results seem to indicate that the suppression resulting from opposite motion in the two eyes is a general phenomenon that it distinct from any mechanisms responsible for detecting motion through depth. The evidence for this is that the suppression holds for both horizontal and vertical motions in both the VEP and psychophysics, whereas anything to do with motion in depth shouldn't hold for vertical motion because human eyes are separated horizontally. Surprisingly (at least to me), the suppression also seemed to require a robust reference to be

present. The infant data seemed to indicate that the surround effects were not yet present at 5 months.

I found the experiments well-done, I'm confident that the data are sound, and I do think they yielded informative results. The manuscript was also well-written throughout, and the figures were nice and well thought out.

I did feel like it was a bit of a struggle to follow the story. Part of this might be structural, although I must admit that there is nothing specific I can point out about this. Part of this might also be some simple things about terminology and adding some more annotations or legendary information to the figures.

I have two suggestions for perhaps improving the clarity. First, I suggest defining some functional terms for the moving and reference parts of the stimulus. For the moving parts, perhaps "full cue" (as is used late in the manuscript on page 17), "IOVD-uncorrelated" (IOVD-u), and "IOVD-anticorrelated" (IOVD-a) or something similar would work. For the reference parts, perhaps "full reference", "noise reference", and "no reference"?

The second suggestion is that legends be added to the data figures using (abbreviated) versions of the functional terms suggested above. This might require, for example, moving the topographic maps to the supplementary materials (as pretty as they are, they all look the same to me, except for the infant map in Figure 7, and of course the "What the hell???" map in middle row of Figure 6. It might also be worth considering breaking Figures 1 and 6 up into two figures each, one for the moving part manipulation and one for the reference manipulation.

The p-values should be expressed as estimated probabilities throughout (rather than as 'equal to' in some cases and 'less than some arbitrary value ending in a 5 or a 1' in others).

Abstract first sentence: I suggest "perceived lateral displacements" instead of "retinal image displacements" for admittedly pedantic accuracy.

Page 3, bottom: Here and elsewhere, the verb "comprise" is used in exactly the way I was always taught not to use it: as a synonym for "compose".

Figure 1: This figure (which is great) could also be used to define the more functional terminology suggested above, which could then be incorporated in legends for the data figures. Also, perhaps it would be possible to tie the colors used in Figure 1 to those used in the data figures. This might require some restructuring of the data figures, and may or may not be worth it.

Oh, also for all the forthcoming figure legends (haha), "0" and "pi" could be used as shorthand to denote the interocular motion phase (following in the long tradition of auditory psychophysics).

Oh, oh, one final suggestion. It might be useful to look at the fits as data points in parameter space (instead of or in addition to doing t-tests on individual parameters). So, for example, the dark orange curve in figure 1A would plot as a single point in a space with the dimensions of R_{max} , n , and $d50$ (b was presumably fixed at 0). This would be a very compact way to represent each Naka-Rushton function visually, and might lead to some useful data visualization.

Reviewers' comments:

Reviewer #1 (Remarks to the Author):

Summary:

Kohler, Meredith & Norcia used Steady-State Visual Evoked Potentials (SSVEPs), to making direct neural measurements of the divergent responses to different forms of suprathreshold motion in human adults and 5-month-old infants. They report that:

- 1) Inter-ocular differences in retinal image motion generated suppressed response functions and correspondingly elevated perceptual thresholds, compared to motion matched between the two eyes.
- 2) This suppression was of equal strength for horizontal and vertical motion and therefore not specific to the perception of motion-in-depth.
- 3) Suppression is strongly dependent on the presence of spatial references in the image, and
- 4) Suppression is very immature in infants.

They conclude “suppression appears to be the manifestation of a succession of spatial and inter-ocular opponency operations that occur at an intermediate processing stage either before or in parallel with the extraction of motion-in-depth.”

Evaluation:

This paper represents a heroic set of 8 experiments; 5 SSVEP experiments and 2 psychophysical experiments in adults ($N = 59$), as well as SSVEP experiments in infants ($N = 22$). The experimental methods and analyses are sensitive and sophisticated, and the results are convincing, interesting and important to our understanding of motion in depth and perceptual suppression.

That said, I suspect that said, the multiplicity of experimental conditions (both targets and references), SSVEP components analyzed (F, 2F, 4F) and different cues to motion in depth (MID) may be confusing to non-specialized readers – indeed, I found them difficult to keep track of, and found myself having to re-read sections and going back to figure 1. One suggestion that may help the reader is to add a column to Figures 2 (and maybe Figs 3 and 4 also) replicating the bottom panel of Fig. 1. A legend in the figure for the different colors in the each panel would also be helpful, rather than having to refer to the long figure caption to determine what conditions each curve refers to. As an aside, the panels showing scalp topography get a lot of space, but almost no discussion. I wonder whether they might be better placed in Supplemental materials.

Response: We agree that this is a rich dataset and that it is important to minimize confusion for non-specialist readers. To that end, we have taken the reviewer’s suggestions to heart. We have modified Figure 1 and added a legend and stimulus icons to Figures 2, 4, 6 and 7 that directly associate the data colors with the behavior of the dots in the test and reference bands. We have also, as suggested, moved the scalp topographies to Supplementary Figure 2.

I applaud the authors for not ‘slicing the salami’ – and this kind of detailed analysis will be of great interest to a specialized vision science audience. However, for a general audience, it may be more helpful to strip the paper down to simply focus on the most critical conditions that highlight the main conclusions.

Response: Rather than removing substantial portions of the results, we have focused on clarifying the presentation and interpretation of the results and on constructing a narrative that non-specialists should be able to follow. This, combined with the changes to the figures, should make the paper accessible to a wider audience.

Some specific Issues:

Introduction: “Here we systematically explore the importance of clearly defined references on neural responses to motion and disparity using minimally complex scene structures comprised of regions defined by motion, disparity or both.”

While the current work goes well beyond, it seems reasonable to reference previous work on this and closely related topics (e.g. Zemon & Vance, PNAS, 1982; Steinman et al., Vision Research, 1985; Visual Neuroscience 1992).

Response: We appreciate the suggestion, and now cite these references in the Introduction.

Results: Page 9 of the review PDF (actual page numbers would be helpful) – the Paragraph beginning “In the first psychophysical experiment . . . ““(see Figure 4)” should be “(see Figure 3)”.

Response: We appreciate that the reviewer caught this mistake. We have corrected the mistake and added page numbers to the document.

The next sentence asks the reader to “(compare dark green and orange in Figure 2 with Figure 4A, B”. That's hard to do since they're on separate pages and there are a lot of curves in Fig. 2 - why not just give the amplitude differences in this expt (Fig 3 - NOT 4) vs those in the previous (Fig 2).

Response: This sentence was somewhat clumsy and unnecessarily confusing. We are merely pointing out that the EEG data collected during the psychophysical experiment over the near threshold range, follow the same pattern seen in the other experiments that were run over the suprathreshold range. We have revised the sentence to make this clearer.

Figure 3 and 4 – consider adding a column replicating the bottom panel of Fig. 1 showing the various conditions.

Response: Our modifications to Figures 2, 4, 6 and 7 now include stimulus icons. This should cover the reviewer’s concern.

Reviewer #2 (Remarks to the Author):

The present study uses steady-state visual evoked potentials (SSVEPs) to measure the neural

correlates of processing motion in depth. The main finding is that inter-ocular differences in retinal image motion resulted in attenuated neural response functions (as indexed by SSVEP amplitudes) that were accompanied by elevated perceptual thresholds, relative to motion that was matched between the two eyes.

This study entails a rich data set - electrophysiological and behavioral measurements across 8 experiments, including an infant study, and a thorough analysis of the data. However, many of the measurements used are not clearly motivated and it remained elusive to me why certain measurements of the SSVEP signal were taken into account for one interpretation or another. Furthermore, many of the statistical results ride along the p-curve - many of the p values reported are at or nearby 0.05, suggesting that the results are to be taken with caution, and the study was overall underpowered. Finally, the infant study seemed like it was simply attached to the adult study - again with no clear theoretical background or motivation. Thus, in a revision the authors should make sure to 1) induce a larger sample to power their statistical analysis appropriately, 2) provide clear theoretical background in what their measurements indicate and why they used them, 3) clarify the motivation for the additional infant experiment. Below please find more detailed points.

1) Before describing the results in detail, it would be useful to go over the experimental design and general aspects of the SSVEP measurements. In particular, why would SSVEP measurements in this case be a reasonable neural correlate of motion processing? What does it tell us beyond the behavioral thresholds? Second, why did the authors focus on second harmonics instead of first, and why did they later on in some of their analysis include 4F as well? How can we interpret different harmonics of the SSVEP? The authors often use this approach in their papers, but it would be good to clearly state what it means in this particular experiment if the responses of different harmonics differ, and how one can interpret them (ideally, a priori and hypothesis-driven, not post-hoc).

Response: SSVEPs and VEPs in general have long been used to study 2D motion processing. For example, direction-specific adaptation has been used to show that SSVEPs tap direction-selective neurons (Ales et al., 2009, *JVision*; Clark et al., 2014, *NatNeurosci*). These papers are now mentioned in the Introduction as examples. Additionally, in our response to the reviewer's question about the motivation for the developmental study (point 7), we provide an additional example from the developmental literature.

An advantage of the SSVEP approach is that it allows us to measure both near-threshold and supra-threshold responses. The former allows us to link the results to the psychophysical literature on thresholds and the latter to the primate electrophysiology literature on responses to supra-threshold stimuli. This is particularly relevant in the current work, because the SSVEP approach allows us to discriminate signal-detection models proposed on the basis of threshold psychophysics from the active suppression-based account proposed here. We now mention this motivation in both the Introduction and Discussion.

Unlike traditional Event-Related-Potential peak latency and amplitude measurements, the different harmonics of the SSVEP have *a priori* interpretations based on symmetry considerations in both the stimulus array and in the response spectrum (see Norcia et al.,

2015, *JVision*). Symmetric stimulation (e.g. left-right motion on the retinae) leads to symmetric responses and a response spectrum that is comprised of even harmonics. Asymmetric stimulation (e.g. exchanges between uniform and segmented stimulus arrays) additionally generates odd-harmonics if the brain has successfully encoded the distinction. We present the second harmonic data first because it is directly relatable to the symmetric changes of direction-of-motion on the retinae. We then turn to the first harmonic because it reflects the more complex encoding of the global stimulus configuration (uniform vs segmented). We end on the fourth harmonic for rhetorical reasons, as it is the only part of the evoked response that differentiates vertical and horizontal directions of motion and thus is the only signal we have found that may be relatable to perceptual motion in depth.

We have added text to the Introduction related to the first two points and have expanded the interpretation of the different harmonics at the beginning of the Results. As mentioned, the material we have added to address the motivation for the developmental study (point 7), also helps address the first point on VEPs as measures of directional mechanisms.

2) Figure 1 displays the different conditions but it was quite difficult to parse. Is there a way to simplify the figure? Also, I would like to see one example of an actual stimulus display used (not just the schematic illustration). In particular for the measurement of SSVEPs it's informative for the reader to see the exact stimulus that's driving the SSVEP responses.

Response: We have simplified Figure 1 and added icons to Figures 2, 4, 6 and 7 that should help tie together the display and the measured responses. In addition to the schematics shown in the paper, we have provided movies of the actual displays used for the full cue/full reference conditions in Experiment 1, in the Supplementary Material (Supplementary Movies 1-4). Red-blue anaglyph glasses are required to view the disparities in the anti-phase conditions.

3) Please explain why the Naka-Rushton function was used, and whether other functions were tested.

Response: We originally used the Naka-Rushton function as a data reduction technique to increase the power of our statistical analyses by reducing the number of comparisons. It was not intended as a model of the response. In response to point 5, we have now replaced this approach with an assumption-free and data-driven approach to significance testing that is based on paired t-tests of the projected amplitudes, both averaged over the bins and at individual bins (see below). We retain the NR fits for descriptive purposes only.

4) It would be useful to add a few sentences about general experimental approach and measurements in the introduction to facilitate following the Results section.

Response: We have added a brief discussion of the general experimental approach and measurements to the beginning of the Results.

5) Many, many of the statistical tests show ambiguous results: they are just significant, or approach significant, or just trending towards significance. That makes many of the results really difficult to interpret. This raises two questions: How was the number of subjects chosen (N=11 -

sees very small to begin with!!) ? It looks as if this study is largely underpowered and should definitely be replicated with a larger set of participants. Second, what are the effect sizes across conditions?

Response: We thank the reviewer for prompting us to examine our approach to significance testing more closely. In the process, we discovered that the curve-fitting approach was obscuring the magnitude of many of our measured effects because responses that are weak and close to the noise floor are difficult to fit reliably despite indicating large experimental effects. As an alternative, we adopted an assumption-free and data-driven approach in which the vector defined by the real and imaginary part of each individual subject's response is projected onto the vector average. Paired *t*-tests could then be run on the projected vector amplitudes across subjects, for each displacement and on averages across displacements as a summary statistic. This approach has much better statistical power.

To address the sample size concern, we have added 19 adult sessions and 8 infant sessions, bringing the sample size to 15 across all adult experiments (5 EEG experiments and 2 combined EEG and psychophysics experiments) and to 22 in the infant experiment.

We now report effect size in terms of Cohen's *D*, with all other statistical results, in tables S1-S8 in the Supplementary Materials. Note that for the clear majority of our tests, the effect size (Cohen's *d*) is "large". We are thus well-powered with sample sizes of 15. For example, Cohen's *d* = 1 indicates a power of 0.95 at $p = 0.05$ with $n=15$.

6) When I followed the experimental set-up correctly it seems the stimulus input is not matched across the conditions that are later compared in the SSVEP signal. Some stimuli move, some don't. These differences will have large effects on the SSVEP amplitudes, just because of the driving stimulus, so independent of what the authors are interested in. If this is the case, differences in amplitude are difficult (if not impossible) to be interpreted as differences in perception / cognitive functions. Please clarify how this was dealt with.

Response: It is important to note that the monocular apparent motion in the test bands of each stimulus condition was the same across all conditions. Two factors (plus stimulus orientation) vary among conditions: one is the interocular phase of the test bands and the other is the interocular correlation (of either the test bands or the reference bands). All time-locking used for the SSVEP analysis relates directly to the frequency of the monocular motion (2 Hz). Changing the interocular phase in the test bands produces effects that are purely binocular modulations of the monocular responses as this is the only factor that varies across condition and because inter-ocular phase can only be computed after the site of binocular combination. The interocular correlation manipulations also create specifically binocular modulations of the monocular motion response because these relationships can only be computed after the point of binocular combination. When the interocular correlation manipulation is applied to the moving regions, it modulates the availability of the CDOT cue. When it is applied to the reference bands, the response changes reflect the modulatory influence of the reference on the monocular/binocular response generated by the test bands. We thank the reviewer for highlighting the

importance of clearly stating these points, and have added similar language at the beginning of the results when we refer to the stimulus diagram.

7) It is unclear to me why the study on infants was included here. No (or very little) theoretical background is provided why infants could help understanding the processes of motion perception. At the same time, this study is not framed as a developmental paper. Thus, as the paper currently is written the infant data seems almost arbitrarily attached as yet another experiments without really providing new insights for the main question of interest.

Response: We agree that we failed to properly motivate the inclusion of the infant experiment, and have now provided this motivation in the Introduction. We now motivate the inclusion of the infants based on what we currently know about the development of relative disparity processing and what we know about the development of binocular motion mechanisms. A short version of what follows has been added to the Introduction.

Recently, we have identified selective immaturities in infant disparity processing (Norcia et al., 2017, *JNeurosci*) using dynamic random dot stimuli that moved in depth. Many years ago, we and others showed that full maturation of primate motion mechanisms is strongly dependent on correlated binocular input during development (for a review see Norcia, 1996, *Eye*). Infant human and non-human primates display pronounced asymmetries of their oculomotor responses to monocular nasalward vs temporalward motion and corresponding asymmetries in their monocular motion VEP responses. This “Developmental Motion Asymmetry” resides in a specifically binocular motion mechanism that is qualitatively immature in infant humans and macaques (Norcia et al., 1991 *Invest Ophthalmol Vis Sci*; Brown et al., 1998 *VisionRes*). It was thus natural for the current study to study the case of binocular stimuli that have both motion and disparity cues. We replicate and expand on the previous results, by demonstrating a binocular immaturity that has two distinct components (1) absence of anti-phase suppression in infants and (2) absence of a reference effect. We find it particularly striking that the pattern of results in the infants can be recapitulated in the adults by removing reference cues. This result further reinforces the important role for reference cues in binocular motion processing, in addition to providing new information about the development of both relative motion relative disparity processing in infants.

8) Methods:

- Applying the same filter twice does not guarantee that that power frequencies outside the filter range are eliminated. Different filters reduce difference frequencies, but the cut-off is never perfect - also not when applying the same filter twice. Further, since SSVEPs are the main focus of the analysis, it seems filtering is not necessary at all.

Response: The band-pass filter provided by the vendor (EGI) was relatively ineffective in removing large DC offsets. Double filtering was used because it gave us better rejection of slow drifts and allowed us to apply a stricter artifact rejection threshold in our preprocessing. It is true that applying the same filter twice does not guarantee that frequencies outside the filter range are eliminated, but it does reduce them. Running the filter twice did not materially affect the SSVEP results.

Reviewer #3 (Remarks to the Author):

I have read manuscript #.

The authors conducted a series of VEP experiments to address a confusing state of affairs which is that, on the one hand, opposite horizontal motion in the two eyes creates a percept of motion through depth (obviously important motion to be able to detect) while, on the other hand, opposing motion in the two eyes also results a reduction for motion sensitivity. The latter has always seemed completely at odds with the necessity of the former.

The authors manipulated two main things in the stimulus: the degree to which disparity cues should have been stimulated (normal moving dots vs. dots that were either un- or anti-correlated across the eyeballs) and the degree to which the moving portions of the stimulus had a reference (static, fixed disparity reference vs. either dynamic uncorrelated noise or no reference).

In addition to the main VEP experiments done in adults, VEP data were collected in 5-month-old infants, and psychophysical data were collected in adults.

Taken together, the results seem to indicate that the suppression resulting from opposite motion in the two eyes is a general phenomenon that is distinct from any mechanisms responsible for detecting motion through depth. The evidence for this is that the suppression holds for both horizontal and vertical motions in both the VEP and psychophysics, whereas anything to do with motion in depth shouldn't hold for vertical motion because human eyes are separated horizontally. Surprisingly (at least to me), the suppression also seemed to require a robust reference to be present. The infant data seemed to indicate that the surround effects were not yet present at 5 months.

I found the experiments well-done, I'm confident that the data are sound, and I do think they yielded informative results. The manuscript was also well-written throughout, and the figures were nice and well thought out.

I did feel like it was a bit of a struggle to follow the story. Part of this might be structural, although I must admit that there is nothing specific I can point out about this. Part of this might also be some simple things about terminology and adding some more annotations or legendary information to the figures.

Response: Reviewer #1 also pointed out the need for structural changes and updates to the figure, and we have made several changes to address their comments and yours.

I have two suggestions for perhaps improving the clarity. First, I suggest defining some functional terms for the moving and reference parts of the stimulus. For the moving parts, perhaps "full cue" (as is used late in the manuscript on page 17), "IOVD-uncorrelated" (IOVD-u), and "IOVD-anticorrelated" (IOVD-a) or something similar would work. For the reference parts, perhaps "full reference", "noise reference", and "no reference"?

Response: We have implemented this useful suggestion.

The second suggestion is that legends be added to the data figures using (abbreviated) versions of the functional terms suggested above. This might require, for example, moving the topographic maps to the supplementary materials (as pretty as they are, they all look the same to me, except for the infant map in Figure 7, and of course the “What the hell???” map in middle row of Figure 6. It might also be worth considering breaking Figures 1 and 6 up into two figures each, one for the moving part manipulation and one for the reference manipulation.

Response: We have added legends with icons to the figure as suggested, and moved the topo maps to the supplementary materials. The strange topo map for *IF* in Experiment 3 looks a little more reasonable with the additional subjects added (see Supplementary Figure 2), but it is important to point out that a strange map is not surprising given that none of the conditions produced measurable *IF* signal in this experiment.

The p-values should be expressed as estimated probabilities throughout (rather than as ‘equal to’ in some cases and ‘less than some arbitrary value ending in a 5 or a 1’ in others).

Response: We now report the exact value for all p 's > 0.0001.

Abstract first sentence: I suggest “perceived lateral displacements” instead of “retinal image displacements” for admittedly pedantic accuracy.

Response: We prefer to use retinal image displacements to avoid using “perceive” twice in the same sentence.

Page 3, bottom: Here and elsewhere, the verb “comprise” is used in exactly the way I was always taught not to use it: as a synonym for “compose”.

Response: We have replaced the word comprise throughout the paper.

Figure 1: This figure (which is great) could also be used to define the more functional terminology suggested above, which could then be incorporated in legends for the data figures. Also, perhaps it would be possible to tie the colors used in Figure 1 to those used in the data figures. This might require some restructuring of the data figures, and may or may not be worth it.

Response: We have added the functional terminology to Figure 1 and incorporated it in the caption of the data figures.

Oh, also for all the forthcoming figure legends (haha), “0” and “pi” could be used as shorthand to denote the interocular motion phase (following in the long tradition of auditory psychophysics).

Response: This is a useful suggestion when text figure legends are used, but since we are now using icon legends in Figures 2, 4, 6 and 7, we have opted not to implement the “0” and “pi” shorthand.

Oh, oh, one final suggestion. It might be useful to look at the fits as data points in parameter space (instead of or in addition to doing t-tests on individual parameters). So, for example, the dark orange curve in figure 1A would plot as a single point in a space with the dimensions of

R_{max} , n , and $d50$ (b was presumably fixed at 0). This would be a very compact way to represent each Naka-Rushton function visually, and might lead to some useful data visualization.

Response: Given that we are no longer using the Naka-Rushton function as a data reduction technique (see our response to Reviewer #2) we can no longer implement this otherwise useful suggestion.

REVIEWERS' COMMENTS:

Reviewer #1 (Remarks to the Author):

The revised manuscript, while still presenting a rather complex dataset , is much improved by the new figures, analyses and narrative.

The revised version addresses all the issues that I raised in the original review and the results make a convincing, interesting and important contribution to our understanding of motion in depth and perceptual suppression.

Reviewer #2 (Remarks to the Author):

The authors have addressed all my previous concerns.

Reviewer #3 (Remarks to the Author):

I am completely satisfied with the authors' responses to my comments. Thank you.